# Clinical Landscape of PARP Inhibitors in Ovarian Cancer: Molecular Mechanisms and Clues to Overcome Resistance

**DOI:** 10.3390/cancers14102504

**Published:** 2022-05-19

**Authors:** Satoru Kyo, Kosuke Kanno, Masahiro Takakura, Hitomi Yamashita, Masako Ishikawa, Tomoka Ishibashi, Seiya Sato, Kentaro Nakayama

**Affiliations:** 1Department of Obstetrics and Gynecology, Shimane University Faculty of Medicine, Izumo 693-8501, Japan; kanno39@med.shimane-u.ac.jp (K.K.); meme1103@med.shimane-u.ac.jp (H.Y.); m-ishi@med.shimane-u.ac.jp (M.I.); tomoka@med.shimane-u.ac.jp (T.I.); sato_seiya9534@yahoo.co.jp (S.S.); kn88@med.shimane-u.ac.jp (K.N.); 2Department of Obstetrics and Gynecology, Kanazawa Medical University, Kanazawa 920-0293, Japan; takakura@kanazawa-med.ac.jp

**Keywords:** ovarian cancer, PARP inhibitor, maintenance therapy, homologous recombination deficiency, replication fork, DNA damage response, cGAS/STING, cytosolic immunity

## Abstract

**Simple Summary:**

Recent development of maintenance therapy using PARP inhibitors in ovarian cancer has led to a significant improvement in survival rates. However, resistance to these inhibitors can occur in patients, causing disease progression or relapse. Consequently, novel treatment strategies are urgently needed to overcome this resistance. This review article focuses on the precise molecular mechanisms by which PARP inhibitors exert their antitumor effects, as well as how they elicit resistance, in order to gain insight into novel therapeutic approaches to overcome PARP inhibitor resistance in ovarian cancer.

**Abstract:**

The survival of patients with advanced or recurrent ovarian cancer has improved tremendously in the past decade, mainly due to the establishment of maintenance therapy with poly (ADP-ribose) polymerase (PARP) inhibitors (PARPis) after conservative chemotherapies. Despite their superior efficacy, resistance to PARPis has been reported, and patients with resistance have a much worse prognosis. Therefore, the development of novel treatment strategies to overcome PARPi resistance is urgently needed. The present review article focuses on the molecular mechanisms of how PARPis exert cytotoxic effects on cancer cells through DNA repair processes, especially the genetic background and tumor microenvironment favored by PARPis. Furthermore, currently available information on PARPi resistance mechanisms is introduced and discussed to develop a novel therapeutic approach against them.

## 1. Introduction

The concept of maintenance therapy after platinum-based chemotherapy has rapidly developed in the first-line setting of advanced epithelial ovarian, tubal, and peritoneal cancers (collectively referred to as ovarian cancer) or after recurrence, in which poly (ADP-ribose) polymerase inhibitors (PARPis) hold important roles, providing prolonged progression-free survival (PFS) and overall survival (OS) compared to conservative chemotherapy alone [1,2,3,4,5,6]. Nevertheless, a considerable proportion of patients still show disease progression or recurrence after maintenance therapies, and the prognosis is much worse, although several approaches, including repeated PARPi treatment, have been attempted to treat such patients [7]. To predict or protect against disease progression or the recurrence of ovarian cancers after maintenance therapy, the key is to develop a profound understanding of the molecular mechanisms by which PARPis exert specific anti-tumor effects, distinct from those of conservative chemotherapy. The present review article focuses on those mechanisms, mentioning the genetic background and tumor microenvironment (TME) of ovarian cancers before, during, and after PARPi treatment. The importance of combination treatment strategies based on an understanding of complex molecular rationales is also stressed. Finally, methods to overcome PARPi resistance are discussed based on the currently available information on the detailed molecular mechanisms that cause resistance.

## 2. Two Distinct Therapeutic Mechanisms of PARPis

### 2.1. Enzymatic Inhibition

As a family member of enzymes composed of various isoforms, including PARP1/2 [8], PARP plays am essential role in the base excision repair (BER) of single-strand DNA breaks (SSBs). The founding family member, PARP1, has been the most extensively studied. It binds to the sites of SSBs via the zinc-finger DNA binding domains and catalyzes poly-ADP-ribosylation (PARylation) to proteins that repair SSBs using their enzymatic activity [9,10]. Poly-ADP-ribose (PAR) has a negative charge and serves as a platform to recruit proteins that repair SSBs with a positive charge, resulting in the efficient repair of SSBs. It is a major principle that PARP1 promotes repair of SSBs as an enzyme.

The enzymatic activity of PARP1 is targeted by PARPis, inhibiting PARylation. Eventually, PARPis disturb repair of SSBs, leading to the accumulation of double-strand DNA breaks (DSBs) via subsequent DNA replications. Such DSBs are repaired mainly by specific repair pathways, named homologous recombination repair (HR), a mechanism of DSB repair using sister chromatids as a template [11], although there is another machinery of DSB repair, non-homologous end-joining (NHEJ), that is used when HR is decreased. When cancer cells have HR defects (homologous recombination defect, HRD), they suffer from accumulated DSBs leading to increased genetic and chromosomal instability, finally resulting in cell death. This is mechanistic rationale, by which PARPis exert cytotoxic activity, especially in the presence of HRD, is called synthetic lethality [12,13].

### 2.2. PARP Trapping

Furthermore, PARPis exert an alternative effect on PARP1. Of note, PARP1 induces PARylation not only to proteins that repair SSBs, but also to PARP1 itself. The PARylation of PARP1 itself induces the release from DNA lesions due to charge repulsion, because of the negative charge of DNA. Thus, PARP1 leaves DNA lesions after PARylation by itself. Subsequently, PARPis inhibit the PARylation of PARP1 and, therefore, prevent PARP1 from leaving DNA lesions, allowing PARP1 to be continuously trapped on the DNA (PARP-trapping) [14]. This PARP-DNA complex interferes with DNA replication by stalling the replication fork and thereby functions as a so-called “DNA poison”, which confers collapse of the replication fork and cell death [13,15]. This is an alternative mechanism by which PARPis exert cytotoxic activity. Various repair mechanisms, including HR, are mobilized to eliminate such poisonous complexes. However, the defects of repair mechanisms, such as HRD, allow continuous PARP-DNA complexes, promoting defective DNA replication and cell death.

Thus, two distinct types of PARPi actions, (1) enzymatic inhibition for synthetic lethality and/or (2) PARP trapping for fork collapse and cell death, occur in the context of maintenance therapy. It is important to determine which type of mechanism is clinically more active for cytotoxic activity. In previous in vitro studies [14,16], cancer cells were treated with various concentrations of PARPis, and the statuses of PARylation and PARP trapping were quantified and compared with cytotoxic efficacy. Overall, PARP trapping required higher concentrations of PARPis than enzymatic inhibition in most cell types, and cytotoxic activity was more dependent on PARP trapping than on enzymatic inhibition for PARylation; most cell types exhibited resistance to PARPis in lower concentrations that were sufficient for complete inhibition of PARylation, but insufficient for PARP trapping [14,16]. Therefore, PARP trapping appears to play predominant roles in cytotoxicity, especially at higher concentrations.

## 3. HRD as a Predictive Marker for the Efficacy and Survival Benefit of PARPis

The HR repair mechanism (HRR) involves multiple proteins, and germline *BRCA1/2* mutations are the most well-known genetic component of HRD in ovarian cancer, accounting for 10–15% of cases, especially for high-grade serous carcinoma (HGSOC), in which they account for 20–30% of patients [17,18,19]. Other causes of HRD include somatic *BRCA1/2* mutations, accounting for 1/3–1/2 of the germline *BRCA1/2* mutations [20], promoter silencing of *BRCA1* (approximately 10% of HGSOC) [17], or mutations of other genes involved in HRR. In the presence of HRD, PARPis are expected to show greater cytotoxic effects via enhanced synthetic lethality and PARP trapping.

Recent clinical trials with PARPis showed that the benefit with the highest clinical relevance was observed in *BRCA*-mutated or HRD cohorts [1,2,3,4], and, therefore, *BRCA*-mutation and HRD status have been used clinically as companions to predict the efficacy of PARPis. The myChoice^®^ CDx is the first and only FDA-approved tumor test that detects *BRCA1/2* variants and/or scores HRD status [21]. The scoring is determined by assessing genomic instability with the following critical biomarkers: loss of heterozygosity (LOH), telomeric allelic imbalance, and large-scale state (defined as chromosomal breakage that generates 10 Mb or larger DNA fragments), based on the concept that impaired HR causes accumulation of DSBs and induces genetic and chromosomal instability. The sum of the above three biomarkers was calculated to represent genomic instability status (GIS) or an HRD score and used as a surrogate measure for genetic instability, in which a threshold of ≥ 42 or *BRCA* mutation was considered HRD-positive. Using this system and other candidate biomarkers, including mutations in other HRR genes (including *BR IP1, CDK12, RAD54L, RAD51B, ATM, FANCA, FANCD2, FANCL, RAD51C, RAD52,* and *XRCC*) and *BRCA1* promoter methylation, tumor samples from olaparib maintenance monotherapy (study 19 [22]) were analyzed [23]. Most patients with *BRCA* mutations, and almost all patients with *BRCA1* methylation, were identified as myriad HRD-positive, while less than half of patients with mutations in other HRR genes were identified as HRD-positive [23]. Patients with myriad *BRCA*-mutated tumors exhibited the greatest survival benefit from olaparib (progression-free survival, or PFS, hazard ratio, or HR, of 0.17, with a 95% confidence interval, or CI, of 0.09–0.30), while those with HRD-positive status showed a comparable survival effect (HR 0.24, 95% CI 0.15–0.39), but patients with *BRCA*-wild type/HRD-positive tumors did not have as high a survival benefit (PFS HR 0.48, 95% CI 0.18–1.27) [23]. Of note is that patients with mutations in other HRR genes identified by gene sequencing had an unexpectedly better survival benefit (PFS HR 0.21, 95% CI 0.04–0.86) [23]. Patients with wild-type *BRCA*/HRD-negative status (PFS HR 0.60, 95% CI 0.31–1.17) or those with wild-type *BRCA*/HRR genes (PFS HR 0.71, 95% CI 0.37–1.35) showed the least benefit from olaparib [23]. 

Thus, *BRCA* mutations and/or HRD-positive status are superior biomarkers of survival benefit from olaparib, on which *BRCA* mutations have the greatest impact. Mutations in HRR genes other than *BRCA* may also have treatment benefit from olaparib. However, there were contradictory results in a recent study [24] analyzing the patient population of the phase III PAOLA-1 study [25], in which mutations in HRR genes were not predictive of PFS benefit from olaparib in combination with bevacizumab, compared with bevacizumab alone. Further analyses with larger patient populations are needed to determine the clinical significance of detecting HRR gene mutations other than of *BRCA* as biomarkers predictive of treatment benefit of not only olaparib, but also other PARPis.

## 4. Dynamics of HRD Status in the Treatment of Ovarian Cancer

The status of HRD is vigorously altered during the treatment of ovarian cancer.

Takaya et al. [26] examined the clonality index and LOH scores using tumor samples of HGSOC obtained at primary debulking surgery (PDS) and interval debulking surgery (IDS) after neoadjuvant chemotherapy (NACT) or secondary debulking surgery (SDS) for recurrent tumors, and found that the clonality index and LOH decreased at IDS compared to before NACT, and then subsequently re-elevated at SDS. These findings indicate that intratumoral heterogeneity decreases after chemotherapy, probably because platinum-sensitive clones disappeared due to the platinum-based chemotherapy, leaving selectively grown chemo-resistant clones (Figure 1). 

Since the LOH score is a component of HRD status, these findings indicate that HRD-positive cells in the primary tumors were likely to be eradicated by chemotherapy, whereas HRD-negative cells were selected and occupied residual tumors. Subsequent re-elevation of the clonality index and LOH score suggests that recurrent tumors are composed of heterogeneous clones, including HRD-positive cells, reminiscent of their aggressive re-growth potential during platinum-free periods. Sokolenko et al. examined the LOH of the *BRCA1* allele of HGSOC with germline *BRCA1* mutation (*gBRCA1* MT) in primary tumors in pre-NACT status, residual tumor at IDS, and recurrent tumor at SDS [27], and they found that all primary tumors had *BRCA1* LOH, whereas retention of the wild-type *BRCA1* allele was detected at IDS. Thereafter, most recurrent tumors exhibited *BRCA1* LOH again, indicating that NACT results in the rapid selection of pre-existing wild-type *BRCA1*-proficient cells, whereas tumor relapses re-acquired *BRCA1* LOH during therapy holidays. A similar observation was reported by Patel et al. [28], in which change in HRD status was measured and compared between paired primary and recurrent HGSOC samples, demonstrating that recurrent tumors took over the HRD status of the primary tumors, even after the successful treatment of primary lesions. Thus, maintenance therapy after the primary treatment of HRD-positive tumors needs to target HRD-positive cells, despite tumor samples at IDS exhibiting an HRD-negative phenotype. Therefore, HRD-negative status at IDS does not precisely predict HRD status in recurrent tumors and should not be considered in the choice of PARPis after IDS.

## 5. The Fate of PARP Trapping: Fork Stabilization or Degradation That Determines the Efficacy of PARPis

As described earlier, PARP trapping on DNA by PARPis blocks DNA replication via stalled replication forks, triggering a series of replication stress responses [29]. Unless stalled forks are successfully repaired by such stress responses, they are extensively degraded by nuclease proteins and undergo irreversible fork collapse, leading to genomic instability and cell death [30,31].

A series of stress responses triggered by stalled replication forks are illustrated in Figure 2.

A stalled replication fork is very unstable and is exposed to the risk of fork collapse. There are rescue processes for stalled replication forks [32]. After fork stalling, ssDNAs are generated by polymerase-helicase uncoupling in both lagging and leading strands. These ssDNA regions are coated by replication protein A (RPA) to prevent the formation of secondary structure [33]. Thereafter, the ssDNA-RPA complex activates the replication checkpoint via ataxia-telangiectasia-mutated protein kinase (ATR) and the subsequent CHK1 and Wee1 activation [32,34], which arrests the cell cycle to allow the time necessary to relieve the stalled fork and protect against fork collapse, and, therefore, composes a part of DNA damage responses (DDRs).

Subsequently, RPA was replaced by RAD51, which promotes fork reversal, a phenomenon to bypass and unwind the stalled fork [35]. This reversal is achieved by nascent lagging strands caused by blocked leading strands, which serve as an alternative template for leading strand DNA synthesis toward the opposite direction of fork progression, which has protective and stabilizing effects on stalled forks. It is important to consider why this reversal phenomenon functions to stabilize stalled forks. Fork progression across template DNA lesions, such as SSBs, induces RecQ1-dependent DSB generation as the fork progresses, which finally results in fork collapse [36]. The reversal of the fork protects it from colliding with the DNA lesions, thus avoiding DSB-induced fork collapse. Furthermore, fork reversal towards the opposite direction to replication impediments may allow additional time and room sufficient for the repair machineries to remove them [37].

Stalled replication forks are characterized as exposed DNA ends, which render them susceptible to a variety of nucleases including MRE11 and EXO1 [38]. These nuclease activities destabilize not only stalled forks, but also reversal forks, thereby threatening fork collapse. The BRCA2-RAD51 axis plays pivotal roles in preventing such deleterious fork degradation [39]. As a result, BRCA2 is likely to localize to the stalled forks and promote the formation of stable RAD51 nucleoprotein filaments, therefore inhibiting MRE11-medicated fork degradation [40], in which BRCA1 also contributes to fork stabilization. Whereas several RAD51 paralogs, including XRCC2 and XRCC3, are also required for preventing MRE11-medicated fork degradation [41], various factors including RADX counteract with RAD51 nucleoprotein filaments and, thereby, permit degradation of the fork by MRE11 or EXO1 nuclease, leading to genomic instability or cell death [42]. When BRCA1/2 is defective, MRE11 targets unprotected reversed forks and starts fork resection, leading to the formation of ssDNA flaps in the reversed fork [43], which MUS81 cleaves, resulting in the generation and accumulation of DSBs [38]. These nuclease-induced DSBs are repaired by POLD3-dependent DNA synthesis, contributing to restarting replication recovery from cleaved forks via the break-induced replication (BIR) pathway [38]. Many factors are involved in the repair of such DSBs as parts of DNA damage signaling, including ataxia-telangiectasia-mutated protein (ATM).

Recently, spartan (SPRTN), a DNA-dependent metalloprotease, has been reported to be directly involved in the repair of PARP1-DNA complexes [44]. In the body, SPRTN is recruited to and interacts with trapped PARP1 at replication forks, and it facilitates the repair of bulky PARP1-DNA complexes via proteolytic digestion, leading to replication bypass of PARP1-DNA complexes to ensure fork progression, conferring PARPi resistance.

Overall, integrity of fork stabilization mechanisms on PARP trapping plays major roles in determining sensitivity to PARPis. A number of nuclear factors involved in this integrity can, therefore, be potential targets to overcome PARPi resistance. Especially, since some DDRs usually function to counteract fork stabilization, the inhibitors against constituents of DDR (DDRis) may have a potential to promote fork stabilization. The deficiency or inhibition of ATM are potential targets or candidates to improve the efficacy of PARPi [45]. Preclinical studies demonstrated that the ATM inhibitor AZD0156 enhanced the effects of olaparib across a panel of breast, gastric, and lung cancer cell lines, as well as potentiating the efficacy of olaparib in patient-derived xenografts (PDXs) of breast cancers [46]. Phase I studies of AZD0156 are currently ongoing (NCT02588105). The combined inhibition of PARP and ATR was shown to overcome PARPi resistance in PDXs of ovarian cancer [47]. The phase II trial (CAPRI trial) to use the ATR inhibitor ceralasertib (AZD6738) in combination with olaparib for recurrent ovarian cancer (NCT03462342) is ongoing [48], but has reported preliminary results of an overall response rate (ORR) of approximately 50% for the HRD patients with acquired PARPi resistance. The CHK1 inhibitor prexasertib has been combined with olaparib for HGSOC in a phase II study, and an ORR of 29% was reported in *BRCA*-wild type recurrent HGSOC or high-grade endometrioid ovarian carcinoma (HGEOC) patients [49]. The Wee1 inhibitor adavosertib was combined with olaparib for patients with PARPi-resistant ovarian cancer in the phase II EFFORT study, showing an overall RR of 29% in combination vs. 23% for adavosertib alone, and a median PFS of 6.8 months in combination vs. 5.5 months for adavosertib alone [50]. Adavosertib was combined with olaparib for refractory or recurrent ovarian cancer patients with *TP53* and/or *KRAS* mutations in a phase-II OLAPCO trial (NCT02576444) [51], based on the proposed concept that the combination of Wee1 inhibitors and *TP53* loss will theoretically give coordinated efficacy, since Wee1 involves G2/M transition and p53 involves G1/S transition. Encouraging results, with an ORR of 43%, were obtained. The OLAPCO (NCT02576444) [52] and ATARI (NCT04065269) [53] trials are also ongoing for the combination of olaparib with ceralasertib.

## 6. PARP Inhibition Triggers Tumor Immunogenicity

### 6.1. Defective DNA Damage Responses Enhance Tumor Antigenicity

Maintenance of genome integrity through DNA replication is essential for cell growth and survival. To faithfully repair DNA lesions that threaten genome integrity, multiple DDRs are induced [54], some of which are composed of pathways for DNA repair, including BER, mismatch repair (MMR), HRR, and NHEJ. Defects of DDRs drive not only genomic instability, but also genomic mutability due to the inability to properly repair DNAs (Figure 3).

However, all DDR defects do not equally result in genomic mutability. MMR deficiency involves defects of mismatch repair proteins, including MLH1, PMS2, MSH2, and MSH6, that function to repair mismatched base pairs, insertions, and deletions at DNA duplication [55]. Eventually, MMR deficiency is likely to cause increased point mutations rather than large defects or gains of bases, generating the microsatellite instability (MSI) phenotype. It, therefore, provides readily available reservoirs of neoepitopes caused by elevated tumor mutational burden (TMB), as well as elevated tumor neoantigen burden (TNB), both of which lead to increased immunogenicity with enhanced recognition by T cells [56], triggering robust antitumor immunity. Therefore, superior responses to anti-PD1/anti-PD-L1 therapy can be expected in patients with MMR deficiency [57]. A correlation between increased TMB or TNB and superior effects of immune-checkpoint inhibitors (ICIs) has been proven in a variety of tumor types, leading to the tumor-agnostic approval of pembrolizumab for the treatment of advanced solid tumors with TMB [58]. Defects of other types of DDRs can also generate increased TMB. Defects in HR or BER are known to correlate with increased TMB or TNB, accompanying higher levels of tumor-infiltrating lymphocytes [59,60,61,62]. However, there are some conflicting data that HRD status does not result in increased TMB or TNB [63]. In ovarian cancers, genomic analyses showed less than 10 genetic mutations per megabase [64], and TMB or TNB is not likely to occur, whereas other genetic alterations with LOH, gene amplification, and telomeric allelic imbalance, evaluated as HRD scores, are more frequent. In fact, single-agent ICIs have been shown to exhibit less efficacy in ovarian cancer than in other tumor types [65,66,67].

Even though TMB or TNB due to defective DDRs is required for the initial priming of the antitumor immune responses, the scenario is not so simple, and they do not always lead to subsequent responses with T cell accumulation in some tumor types [68]. Some studies showed that the defects in HR can predict responses to ICI, but in a TMB-independent manner [69]. Therefore, understanding of not only tumor antigenicity, but also tumor adjuvanticity, is a key to exploiting the possible linkage of defective DDRs to tumor immunogenicity.

### 6.2. PARPis Activate the cGAS-STING Pathway and Mediate Cytosolic Immunity

The interface between DDR and immunogenicity has recently attracted special attention in the field of immune-oncology, in which the cyclic GMP-AMP synthase-stimulator of interferon genes (cGAS-STING) pathway plays essential roles [70]. This pathway originally functions as a component of cellular host defense responding to pathogenic invasion, such as viral infection. Foreign cytosolic DNAs from invaded pathogens can be detected by the cytosolic DNA sensor, cGAS, which subsequently activates the STING, leading to activation of type I interferon (IFN) signaling, essential for innate immune responses [71]. Recently, the cGAS-STING pathway has been found to be activated by endogenous defects of DDRs [70,72,73,74], including defects of BER, HRR, or NHEJ (Figure 3). First, cGAS can detect tumor-derived DNAs, which are accumulated by defective DDRs. It is important to understand why tumor-derived DNAs accumulate in defective DDRs. For example, defects of HRR (HRD) leave DSBs unrepaired or insufficiently repaired, resulting in stalled replication forks, triggering replication stress responses, in which several nucleases are activated. Subsequently, MUS81 functions to resolve stalled replication forks by DNA end resection, and it can mediate the cleavage of DNA structures at stalled replication forks [75]. The resolution of Holliday junctions is promoted by EXO1 as a result of its exonuclease activity [76]. Thus, these nucleases generate fragmented DNAs as by-products of their nuclease activity, which are transported to the cytosol from the nucleus through passive spilling or active transport to generate cytosolic nucleic acids (cNAs). Generated cNAs are perceived by cGAS via various pattern recognition receptors (PRRs), with which innate immune systems recognize pathogenic microorganisms as non-self. This recognition activates synthase activity, leading to further activation of STING, inducing type I interferons and diverse cell-autonomous immune responses, leading to cytosolic immunity. Activation of cytosolic immunity via the cGAS-STING pathway functions to promote T cell infiltration and turn immunologically “cold” tumors into “hot” tumors.

Other than replication stress responses, there is another factor that triggers cGAS-STING signaling. Mitotic defects caused by inadequately repaired DNAs [77] or telomere dysfunction [78] are known to induce chromosomal instability and generate micronuclei via abnormal chromosome segregation, in which the fragile nuclear envelope of micronuclei is likely to rupture, leading to the release of chromatin and nuclear acid components into the cytosol, triggering cGAS-STING signaling (Figure 3). Thus, both DDR defects and mitotic defects result in the cGAS-STING pathway inducing cytosolic immunity. Consistent with this concept, recent studies have shown a novel aspect of the treatment strategy, that DDR-targeting agents [73,77,79,80] or radiotherapy [76,81] can activate cytosolic immunity via the cGAS-STING pathway. In addition, low levels of the cGAS-STING pathway are linked to a poor prognosis in various tumor types, and established tumor cell lines do not usually exhibit this pathway, indicating that the cGAS-STING pathway plays a role in preventing tumor progression [82]. As one of the representative DDR-targeting agents, PARPis were confirmed to elicit tumor-antitumor immunity via the cGAS-STING pathway [83,84,85,86,87,88]. The landmark study was published by Ding et al., demonstrating that olaparib triggers local and systemic antitumor immunity in a *BRCA*-deficient context via the activation of the cGAS-STING pathway using a genetically engineered mouse model of HGSOC [85]. They showed that olaparib treatment increased the number of intratumoral CD4^+^ and CD8^+^ T cells, as well as enhanced production of IFNγ and TNFα, together with increased contents of dsDNAs and micronuclei in the HRD context, but not in the homologous recombination proficiency (HRP) context, and the resultant activation of STING signaling was responsible for PARPi-induced antitumor immune responses, proven by STING-knockout mice [85]. These findings suggest the essential concept that PARP inhibition leads to STING-dependent antitumor immunity via generated dsDNAs and micronuclei in tumors with HRD.

In the clinical samples, PARPi-induced antitumor immunity via STING-mediated interferon responses was confirmed. In the RIO clinical trial (EudraCT 2014-003319-12), for triple-negative breast cancers treated with rucaparib, gene expression analysis was performed to compare the molecular profiling of tumor samples before and after the treatment with rucaparib [89]. Subsequently, RNA sequencing identified that type I and type II interferon signaling pathways were upregulated in the biopsy samples from patients treated using rucaparib, together with STING1 expression, which was observed only in HRD tumors.

### 6.3. Upregulation of PD-L1 by PARP Inhibition

Another aspect of PARPi-induced antitumor immunity is the modulation of immune checkpoint expression. Using breast cancer cell lines and mouse xenograft models, Jiao et al. demonstrated that PARPis upregulated PD-L1 expression, associated with GSK3β inactivation [90], and that PD-L1 upregulation by PARPis attenuated the therapeutic efficacy of PARPis via tumor-associated immunosuppression, as well as determining that simultaneous inhibition of PARP and PD-L1 increased sensitivity to PARPi therapy. An in vitro experiment using ovarian cancer cell lines also confirmed that PD-L1 expression was upregulated after PARPi treatment through the CHK1 pathway [91] (Figure 3). Similar upregulation of PD-L1 by PARPis was more recently observed in pancreatic cancer cells in vitro and in vivo, in which JAK2/STAT3 signaling was involved [92].

### 6.4. Clinical Application to Combine PARPi with ICIs

The PARPi-induced activation of the cGAS-STING pathway and the upregulation of PD-L1 propose molecular rationales for a combination strategy with ICIs. The phase II study (NCT02484404) with olaparib plus the PD-L1 inhibitor (durvalumab), was performed for platinum-resistant recurrent ovarian cancer, and a response rate (RR) of 15% and stable disease (SD) of 38% yielded a disease control rate (DCR) of 53% [93]. A single-arm phase I-II trial of niraparib plus pembrolizumab in patients with platinum-resistant recurrent ovarian cancer reported complete response (CR) in 5% of patients, partial response (PR) in 13%, and SD in 47%, and the ORRs were consistent, irrespectively of platinum-sensitivity, previous bevacizumab treatment, and *BRCA* or HR status [94]. Of note is that responses in patients with wild-type *BRCA* or HRP were higher than expected with either agent as monotherapy. The phase II trial of a combination of dostarlimab, bevacizumab, and niraparib in patients with platinum-resistant recurrent ovarian cancer showed PR of 18% and SD in 59%, yielding a DCR of 77% [95]. For platinum-sensitive recurrent ovarian cancers, greater clinical benefit was observed. The phase II study of olaparib and durvalumab (MEDIOLA) for patients with *gBRCA*-mutated platinum-sensitive relapsed ovarian cancer reported ORR of 72% and a median PFS of 11.1 months [96]. The phase II study of olaparib, durvalumab, and bevacizumab (MEDIOLA) for patients with non-*gBRCA*-mutated platinum-sensitive recurrent ovarian cancer showed ORRs of 31% and 77% with an olaparib + durvalumab doublet and an olaparib + durvalumab + bevacizumab triplet, respectively, with a median PFS of 5.5 months and 14.7 months, respectively [97]. Many phase II-III studies are currently ongoing to compare the efficacy of PARPi monotherapy and in combination with ICIs, especially in the context of maintenance therapy after 1st-line chemotherapy, and the results of these studies will greatly affect and change the concept of maintenance therapy for ovarian cancer.

## 7. PARPis Can Cooperate with Antiangiogenic Agents

Hypoxia is a particular microenvironment of growing tumors, occurring due to the imbalance between tumor growth and angiogenesis. Decreased levels of HR factors, such as BRCA and RAD51, were observed in cancer cells during hypoxic exposure [98], mainly caused by epigenetic silencing of the *BRCA* promoter or the transcriptional repression of the *RAD51* promoter via an E2F4/p130 complex, a transcription factor induced by hypoxia that occupies the promoter to inhibit access of other transcriptional activators [99], compromising the HR pathway (Figure 4). Therefore, hypoxic conditions are likely to impair the function of HR, providing a TME favored by PARPis. On the contrary, hypoxic conditions provide another aspect for the TME of cancer cells exposed to PARPis. Hypoxia triggers the activation of VEGF expression via hypoxia-inducible factor 1 (HIF1) [100]. A recent study found that VEGF interacts with neuropilins (NRPs), a family of VEFG receptors [101], and VEGF/NRP signaling promotes HR via activation of the Hippo pathway transducers YAP and TAZ, which are critical downstream targets of VEGF signaling and which activate the *RAD51* promoter via the transcription factor TEAD (Figure 4).

Thus, the VEGF-NRP-YAP/TAZ axis plays a pivotal role in the hypoxia-induced activation of HR. Taken together, there are contradictory aspects of HR in hypoxia, in that HR potential is inhibited via decreased BRCA or RAD51 activity, whereas it is activated via the VEGF-NRP-YAP/TAZ axis. Therefore, targeting the VEGF-NRP-YAP/TAZ axis by anti-VEGF/VEGFR inhibitors effectively induces HRD in cancer cells in theory, a TME favored by PARPis. Various anti-VEGF/VEGFR inhibitors are thus challenged to enhance or cooperate in the effects of PARPis in clinical settings. It is of note that PARP inhibition is known to down-regulate BRCA1 and RAD51 expressions via increased occupancy of the promoters by the transcriptional repressor E2F4/p130 [102] (Figure 4). Thus, PARPis may create their favored TME by themselves, and can, therefore, exert enhanced efficacy.

Overall, these findings create the rationale for trying the combination of PARPis with antiangiogenics. Such combination therapy was initially studied in a phase II trial, in which olaparib plus cediranib, a VEGFR inhibitor, was compared to olaparib alone for platinum-sensitive recurrent ovarian cancer [103]. With the addition of cediranib, PFS was significantly improved, especially for *BRCA*-wild-type patients (median 16.5 months vs. 5.7 months, HR 0.32, *p* = 0.008), but not for those with *gBRCA* mutations (median 19.4 months vs. 16.5 months, *p* = 0.16). Subsequently, OS for *BRCA*-wild-type patients was reported to be significantly improved by the addition of cediranib (median 37.8 months vs. 23.0 months, *p* = 0.047) [104]. The important finding is that cediranib exerted additive survival effects in the HRP context, but not HRD, consistent with the above rationale. The AVANOVA2 study also investigated the benefit of adding bevacizumab, a VEGF antibody, to niraparib in platinum-sensitive recurrent ovarian cancer, and it confirmed the additive effect (median PFS 11.9 months vs. 5.5 months, HR 0.35, *p* < 0.0001), irrespective of HRD status, for patients with HRP (HR 0.36) or HRD (0.40) [105]. The encouraging result of this study is that the additive survival effect was more significant in HRP patients than in HRD patients, considering the rationale of this study. The phase III PAOLA trial compared the efficacy of olaparib plus bevacizumab with bevacizumab alone in stage III-IV high-grade ovarian cancer as maintenance after first-line therapy [25], and it demonstrated the significant improvement of PFS by the addition of olaparib (median 22.1 months vs. 16.6 months, HR 0.59, *p* < 0.001), especially for the HRD cohorts (median 37.2 vs. 17.7 months, HR 0.33; 95% CI, 0.25–0.45) and for the HRD without *BRCA* mutations (median 28.1 months vs. 16.6 months, HR 0.43; 95% CI, 0.28–0.66). However, the results of the HRP cohorts, for which the combination strategy was expected to exert a greater additive effect, were disappointing (PFS median 16.9 months vs. 16.0 months, HR 1.00; 95% CI, 0.75–1.35). Furthermore, this study lacked an olaparib-alone arm to identify whether the combination (addition of bevacizumab) improves the outcomes compared to olaparib alone. Another phase III study (NRG-GY004) compared olaparib plus cediranib and olaparib alone to standard platinum-based therapy in recurrent platinum-sensitive ovarian cancer [106]. However, olaparib/cediranib did not improve PFS versus chemotherapy in the overall population (HR 0.86, 95% CI, 0.66–1.10). In *gBRCA*-mutant patients, olaparib/cediranib showed better PFS than chemotherapy (HR 0.55, 95% CI, 0.32–0.94), and also olaparib (HR 0.63, 95% CI, 0.37–1.07). In *gBRCA* wild-type patients, olaparib/cediranib or olaparib did not a show superior effect on PFS (HR 0.97, 95% CI, 0.73–1.30 and HR 1.41, 95% CI, 1.07–1.86, respectively). Several trials are ongoing to evaluate the efficacy of combinations of PARPis and antiangiogenic agents.

## 8. Molecular Factors Involved in PARPi Resistance

A variety of molecular factors involved in PARPi resistance have been reported or proposed using in vitro models, mouse models, and clinical samples. Some of these molecular factors affect HR genes or HRR, while others affect the pharmacokinetics of PARPi. The representative mechanisms and factors are described in the following sections.

### 8.1. Reversion Mutation

The representative PARPi resistance mechanisms include the restoration of HR-mediated DNA repair impairment via the reversion mutation of HR genes. Since PARPis do not target oncogenic driver genes, but rather the function of HR genes, the restoration of function of HR genes is likely to reduce the efficacy of PARPis. The secondary mutations of the *BRCA* gene after treatment with platinum or PARPi have the potential to restore the native open reading frame of the *BRCA* gene [107,108,109], and are therefore named “reversion mutations” (Figure 5 and Figure 6A).

The *BRCA* reversions account for 20–30% of the cases of disease progression after treatment with platinum or PARPis [110]. Reversion mutations include true reversions (to wild-type sequence), as well as second-site reversions, which are usually intragenic deletions, mostly being flanked by short regions (1–6 bp) of DNA sequence microhomology [107,109] (Figure 5). This microhomology is thought to be the “scar” of DNA-repair processes that use the microhomology region to repair DSB, named microhomology-mediated end joining (MMEJ) [111]. Thus, MMEJ, which probably occurs as a result of impaired HR, may be the predominant cause of the reversion, indicating that HRD may be the primary upstream event driving reversion mutation. Although reversion events occur not only in tumors with *BRCA* mutation, but also in other HR genes, including the *RAD51* gene, most reversion mutations occur in *BRCA1/2* [112]. Even where patients had a common pathogenic mutation, the reversion mutations that emerged in each patient were unique [112] (Figure 5). Therefore, no strong propensity was observed for any particular reversion mutation to result from a particular pathogenic mutation. In *BRCA2*, reversion mutations occur in a position-dependent manner, and the N-terminal regions have a “hotspot” of reversion mutations while the C-terminal regions lack them [112]. This indicates that pathogenic mutations in the C-terminal regions are unlikely to induce second-site mutation and are at lower risk of developing resistance via reversion. Splice-site pathogenic and missense mutations in *BRCA1/2* are not likely to cause reversion mutations, whereas truncating mutations cause them more frequently [112]. It is of note that patients with *BRCA* mutations that involve structural variants, including the homozygous deletion of the entire locus, that are essentially resistant to reversion mutations, are likely to show long-term responses to PARPis [113].

### 8.2. BRCA Hypomorphic Proteins

Some variants of BRCA1 have been reported with similar but weaker biological function of the wild-type BRCA1, named hypomorphic variants, which lack some of the entire domain of the protein (Figure 6B). The existence of such hypomorphic variants has been prevalently identified in a PARP-resistant, patient-derived, xenograft (PDX) model [114], suggesting that such hypomorphs play some roles in PARPi-resistance, probably via lesser but functional effects of the protein. Hypomorphic variants are presumed to be generated by gene rearrangements via various mechanisms, such as alternative mRNA splicing or alternative initiation of translation, that can bypass deleterious *gBRCA1* mutation and thereby restore HR function [115,116,117,118]. Whether these hypomorphs can exert biological effects equivalent to wild-type BRCA1, and if they can be related to the levels of PARPi-resistance, should be tested by pre-clinical models, as well as in clinical settings.

### 8.3. Epigenetic Reversion

Promoter silencing of the HR genes, including *BRCA1* and *RAD51C*, has been detected in ovarian cancers associated with HRD phenotypes, and this is one of the major causes of gene inactivation [119] (Figure 6A). De-methylation of these promoters can re-express these genes as part of a process named “epigenetic reversion”, which can potently convert the HRD to the HRP phenotype, thereby exhibiting PARPi resistance. Whole genome analyses of paired biopsy samples in primary and acquired platinum-resistant HGSOC have shown that the loss of *BRCA1* promoter methylation is associated with platinum-resistance [120]. In a PDX model of ovarian cancer lacking HR gene mutations but harboring *RAD51C* promoter methylation, treatment with niraparib induced a significant increase in mRAD51C mRNA, associated with loss of the *RAD51C* promoter methylation, indicating that *RAD51C* methylation can be reversed by the treatment pressure of PARPi, which restored RAD51C expression, causing PARPi resistance [121]. In patients with HGSOC enrolled in the ARIEL2 Part 1 PARPi study [122], zygosity of *BRCA1* methylation was extensively analyzed in archival tumor and pre-treated biopsy samples with a quantitative method [123], and patients with homozygous *BRCA1* methylation showed improved clinical outcomes with rucaparib treatment, compared to those with any methylated *BRCA1* ever, indicating that homologous *BRCA1* methylation is more likely to respond to PARPi, and that heterologous methylation is likely to be associated with resistance. Consistent with this, the PDX model with *RAD51C* methylation showed the loss of *RAD51C* methylation after the treatment with rucaparib or niraparib, in which a single unmethylated copy of *RAD51C* was able to confer PARPi resistance [124]. Thus, methylation zygosity is a potential biomarker to predict PARPi efficacy and, therefore, should be assessed when considering PARPi therapy.

### 8.4. Re-Gaining HR Proficiency under BRCA Mutation

The above three mechanisms of HR-related PARPi resistance are conferred by changes in HR gene status, especially *BRCA1,* that regain HR proficiency. An additional mechanism is proposed to inactivate the *TP53BP1* gene encoding p53-binding protein 1 (53BP1) without changing *BRCA1* mutation status. The 53BP1 acts as the central component of the protein complex known as the 53BP1-Shieldin complex [125] (Figure 6C). In the *BRCA1* mutation setting, a *BRCA1*-independent HR mechanism acts in which PALB2-BRCA2-RAD51 plays an alternative role. The 53BP1-Shieldin complex interferes with this HR mechanism, sequestering the PALB2-BRCA2-RAD51 complex from DNA break sites, and it also plays a role in preventing one of the HR processes, namely nucleolytic DNA end processing, which is an initial process of HR [126,127,128]. Thus, the 53BP-Shieldin complex functions to disturb HR function, inducing HRD. Eventually, mutations in components of the 53BP1-Shieldin complex result in restored HR in the *BRCA1* mutation setting, leading to PARPi resistance, confirmed by a PDX model [114] or a clinical sample on PARPi progression [129].

### 8.5. PARP1 Mutation

Trapped PARP acts as a DNA poison that causes stalled and collapsed replication forks, leading to cell death. Therefore, the existence of PARP1 is indispensable for PARPi efficacy. Consistent with this, a mouse study using genetically engineered embryonic stem cells demonstrated that *PARP1* loss is the critical determinant of PARPi resistance [130] (Figure 6D). A recent mouse study also showed that *PARP* mutations that caused a loss of the function to bind DNA, even in the presence of PARPi, conferred PARPi resistance [131]. Of note is that a *PARP1* mutation (R591C; c.1771C > T) was detected in an ovarian cancer patient with olaparib resistance, with such mutation predicted to disturb PARP trapping [131]. However, the clinical relevance of *PARP1* mutations has not yet been fully clarified in a larger number of patients, and it will, therefore, be important to know the prevalence and potency to affect PARPi efficacy in ovarian cancer patients.

### 8.6. PARG Loss

It is understood that PARP1/2 are enzymes that catalyze PARylation of their target proteins. As mentioned above, they also catalyze PARylation of themselves (called auto-PARylation), facilitating their dissociation from DNA via electrostatic repulsion between PARylation products and DNA, both of which possess negative charges. This is a mechanism of PARPs to detach from DNA after completing SSB repair. In contrast to this, poly (ADP-ribose) glycohydrolase (PARG) exerts counteracting activity, degrading nuclear PAR [132]. Therefore, the loss of PARG function leads to enhanced auto-PARylation, allowing PARP to detach from DNA, thereby suppressing PARP-trapping by PARPis (Figure 6E). Genetic screening with multi-omics analyses of matched PARP-sensitive and PARP-resistant mouse breast tumors with *BRCA2* mutation identified *PARG* loss as a major mechanism of PARPi resistance [133]. An in vitro study using human cancer cells with *BRCA1/2* deficiency demonstrated that chemically-mediated or shRNA-mediated *PARG* inhibition resulted in increased PAR and decreased PARP1 trapping, as well as increased resistance to PARPi [133]. The immunohistochemical analysis of HGSOC samples demonstrated that PARG protein was broadly expressed in most biopsied samples, but PARG-negative areas could be detected in a significant proportion of samples, which were tightly associated with increased levels of PAR [133]. Thus, PARG is an important mediator of the PARPi response, and the loss of PARG may be a biomarker potentially linked to PARPi resistance.

### 8.7. Schlafen11 Loss

Schlafen family member 11 (SLFN11) is a putative DNA/RNA helicase that is recruited to the stressed replication fork that halts the fork proceeding or promotes degradation, independently of the canonical DDR and, therefore, suppresses cell proliferation [134]. Expression of SLFN11 was known to be a predictor of sensitivity to multiple DNA-damaging agents, including platinum agents and PARPis [135,136,137]. To identify the candidate markers of olaparib response, the correlation of gene expression profiling and olaparib responses was examined in various HGSOC cell lines, and SLFN11 expression was confirmed to be the most strongly associated with olaparib responses [138] (Figure 6F). Recently, a retrospective analysis of 110 HGSOC patients was performed from the phase II, randomized, placebo-controlled, olaparib maintenance trial (NCT00753545) [22], in which SLFN11 expression was evaluated [139]. The expression of SLFN11 significantly impacted PFS, in which SLFN11-high patients had a longer PFS of 12.4 months in the olaparib arm, compared to 3.1 months in the placebo arm. In contrast, the SLFN11-low group had a shorter PFS of 6.3 months in the olaparib arm and 5.1 months in the placebo arm. A similar trend also was observed in OS. Thus, high levels of SLFN11 were associated with improved clinical outcomes to olaparib in HGSOC, indicating that SLFN11 can be a candidate predictor of PARPi sensitivity.

### 8.8. ABCB1 Overexpression

One of the crucial drug resistance mechanisms includes the active removal of cytotoxic agents from cancer cells through the overexpression of the ATP-binding cassette (ABC) drug efflux transporter P-glycoprotein ABCB1, known as multidrug resistance 1 (MDR1); various types of anti-cancer agents are substrates of ABC drug efflux [140,141]. Overexpression of ABCB1 has been known to be involved in PARPi resistance in an early study using a genetically engineered mouse model for *BRCA1*-associated breast cancer [142] (Figure 6G). Using clinical samples of the trial that evaluated the efficacy of the olaparib–cediranib combination (NCT02681237) in HGSOC after progression on a PARPi, acquired genomic alterations at PARPi progression were analyzed, and ABCB1 upregulation was observed in approximately 15% of the patients, although other genetic alterations were detected concurrently [143]. Thus, overexpression of ABCB1 may be at least one cause of possible mechanisms of resistance to PARPis, especially in patients heavily treated with chemotherapies prior to PARPi treatment.

## 9. Conclusions

The present review article summarized the genetic background and molecular dynamics of HRD status in treatment with PARPis. Clinicians should pay particular attention to the timing of measuring HRD status during treatment as part of efforts to optimize subsequent therapies. Molecular mechanisms of how PARPis exert cytotoxic effects on stalled replication forks, of which stabilization or collapse is modified by various nuclear factors, that ultimately determine the fate of cells, that is, the sensitivity to PARPi, were described. Some of these factors are strong candidates that should be targeted in strategies to overcome PARPi resistance. Despite this accumulated knowledge, data with clinical samples, especially from post-PARPi tumors, are seriously lacking. Therefore, future clinical trials should be carefully planned to collect samples on enrolment that will be subjected to multi-omics analyses to uncover the factors related to drug responses, hopefully with new sources, such as liquid samples or single cells. These will help develop strategies to overcome PARPi resistance for ovarian cancer patients, especially in later treatment stages.

## Figures and Tables

**Figure 1 cancers-14-02504-f001:**
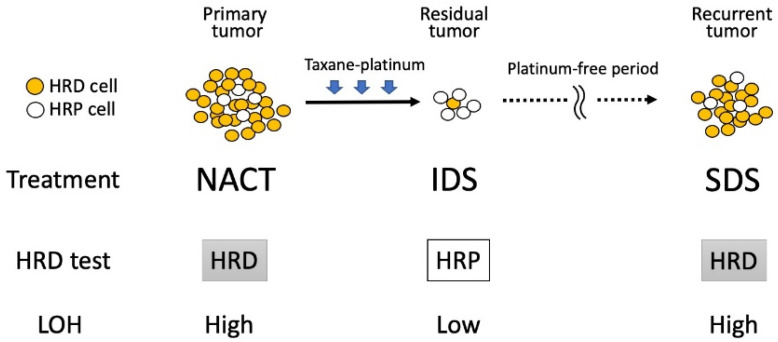
Changes in HRD status during the treatment of ovarian cancer. Even though primary tumors are predominantly composed of HRD-positive cells, they are likely to be eradicated by platinum-based chemotherapy, and residual tumors may be occupied by HRD-negative cells, representing the HRP phenotype. However, after a platinum-free period, remnant HRD-positive cells that have a growth advantage can form recurrent tumors, exhibiting the HRD phenotype. These acronyms are defined as follows: HRD, homologous recombination deficiency; HRP, homologous recombination proficiency; IDS, interval debulking surgery; LOH, loss of heterozygosity; NACT, neoadjuvant chemotherapy; SDS, secondary debulking surgery.

**Figure 2 cancers-14-02504-f002:**
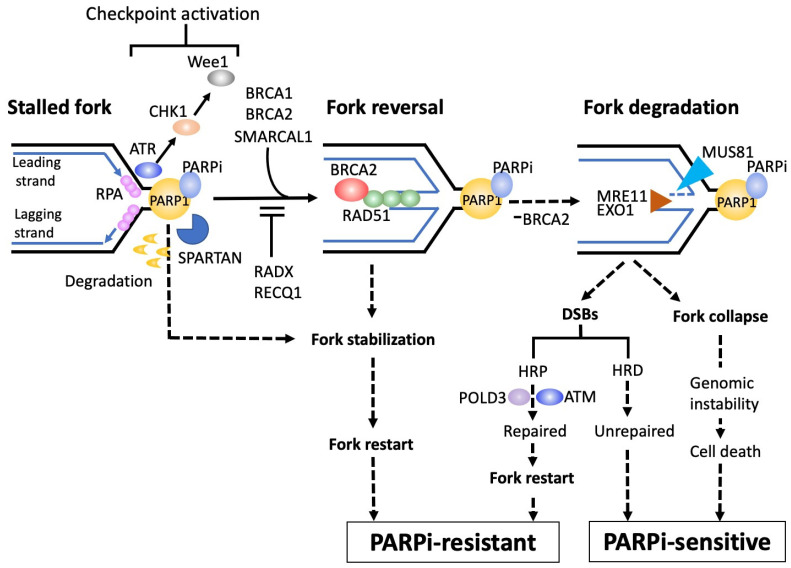
Integrity of fork stabilization on PARP trapping determines PARPi sensitivity. PARP trapping induces stalled replication forks, triggering a series of replication stress responses. After fork stalling, ssDNAs are generated in both lagging and leading strands, and are coated by RPA. The ssDNA-RPA complex then induces activation of the replication checkpoint via ATR and subsequent CHK1 and Wee1 activation, arresting the cell cycle to allow the time necessary to relieve the stalled fork and protect against fork collapse. RPA is then replaced by RAD51, which promotes fork reversal, a phenomenon to bypass and unwind the stalled fork, and the BRCA2-RAD51 axis plays pivotal roles in stabilizing fork reversal. When *BRCA1/2* is defective, MRE11 and/or EXO1 nucleases destabilize stalled forks, resulting in accumulated DSBs or fork collapse, leading to PARPi sensitivity. Spartan (SPRTN) facilitates the repair of bulky PARP1-DNA complexes via proteolytic digestion to ensure fork progression, conferring PARPi resistance. Thus, the integrity of fork stabilization on PARP trapping plays major roles in determining sensitivity to PARPis, and nuclear factors that are involved in this integrity can be potential targets to overcome PARPi resistance. These acronyms are defined as follows: ATM, ataxia-telangiectasia mutated protein kinase; ATR, ataxia-telangiectasia and rad3 related protein kinase; CHK1, checkpoint kinase 1; DSB, double strand break; EXO1, exonuclease 1; HRD, homologous recombination deficiency; HRP, homologous recombination proficiency; MRE11, meiotic recombination 11; MUS81, methyl methanesulfonate and ultraviolet-sensitive gene; POLD3, 3rd subunit of DNA polymerase δ; RADX, RPA-related RAD51-antagonist on X-chromosome; RECQ1, ATP-dependent DNA helicase Q1; RPA, replication protein A; SMARCAL1, SWI/SNF-related, matrix-associated, actin-dependent regulator of chromatin subfamily A-like 1; Wee1, Wee1 tyrosine kinase.

**Figure 3 cancers-14-02504-f003:**
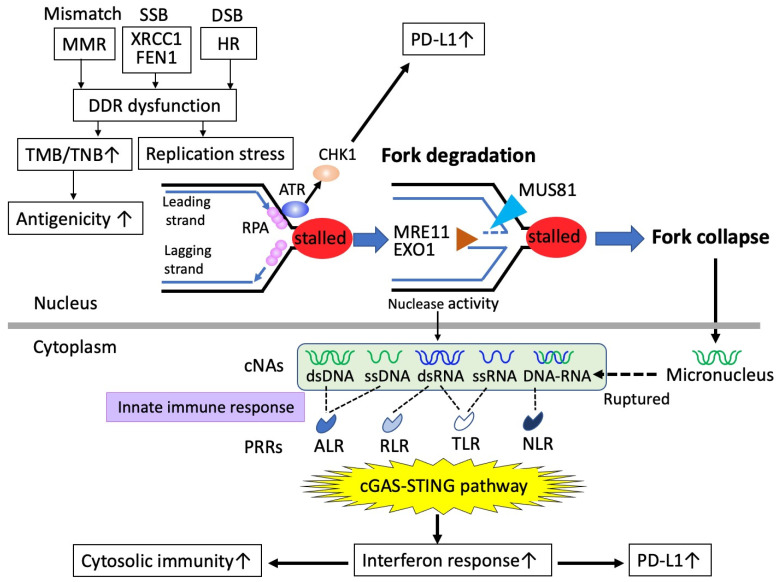
Dysfunction of DDR or PARPis induce the cGAS-STING pathway to activate cytosolic immunity. While some DDR dysfunction leads to increased tumor mutation burden (TMB) or tumor neoantigen burden (TNB), which enhances tumor antigenicity, other DDR dysfunction, such as via PARPis, causes stalled replication forks, triggering ATR/CHK1 activation leading to upregulated PD-L1, and replication stress responses inducing fork degradation via several nucleases, including MRE11, EXO1, and MUS81, result in the generation of cytosolic nuclear acids (cNAs). Fork collapse by the above nucleases induces genetic instability and generates micronuclei, which rupture and become cNAs. These cNAs are recognized by pattern recognition receptors (PRRs) as part of the innate immune response, triggering cGAS-STING signaling to increase cytosolic immunity. These acronyms are defined as follows: ALR, AIM2-like receptors; ATR, ataxia-telangiectasia-mutated protein kinase; cGAS, cyclic GMP-AMP synthase; CHK1, checkpoint kinase 1; DDR, DNA damage response; DSB, double strand break; FEN1, flap endonuclease 1; HR, homologous recombination; MMR, mismatch repair deficiency; PD-L1, programmed cell death ligand-1; RLR, RIG1-like receptors; RPA, replication protein A; SSB, single strand break; TLR, Toll-like receptors; NLR, NOD-like receptors; STING, stimulator of interferon genes; XRCC1, X-ray repair cross-complementing 1.

**Figure 4 cancers-14-02504-f004:**
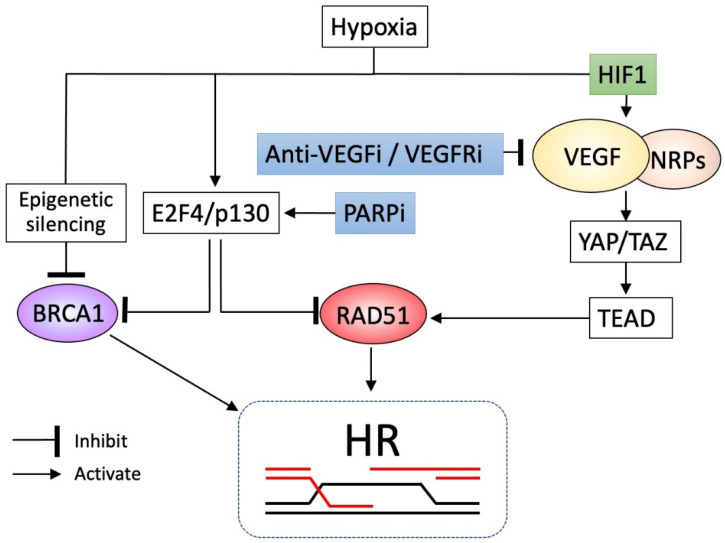
In this figure, PARPis and anti-VEGF/VEGFRi cooperate to induce HRD. The hypoxia-inducible factor (HIF1) increases VEGF expression, which interacts with neuropilins (NRPs), and VEGF/NRP signaling promotes homologous recombination (HR) through the Hippo pathway transducers YAP/TAZ, which activate the *RAD51* promoter via the transcription factor TEAD. In contrast, hypoxia inhibits *RAD51* via transcriptional repressor E2F4/p130 or inhibits *BRCA1* by epigenetic silencing, leading to impaired HR. Thus, hypoxia exhibits contradictory actions, inhibition, or activation of HR. In this context, anti-VEGF agents or VEGFR inhibitors (VEGFRis) suppress VEGF/NRP signaling, causing impaired HR, whereas PARPis inhibit BRCA1 and RAD51 expression via increased formation of the transcriptional repressor E2F4/p130, resulting in enhanced HRD. Thus, PARPis and anti-VEGF/VEGFRi can cooperate to elicit the HRD phenotype to achieve additive or synergistic effects.

**Figure 5 cancers-14-02504-f005:**
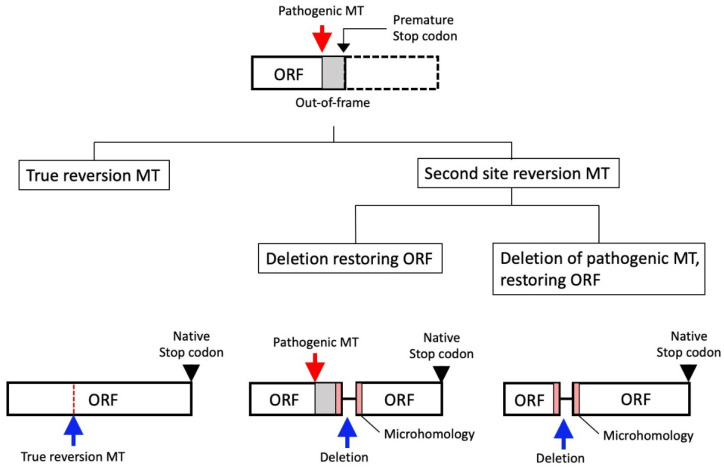
Representative types of reversion mutation. Architecture of the homologous recombination (HR) gene is shown. The HR gene with a pathogenic mutation (MT) and a resultant premature stop codon due to out of frame sequences are illustrated in the upper panel. The normal function of the gene is lost as a result of the MT and resultant premature stop codon. As shown in the lower panel, true reversion MT completely restores the pathogenic MT and open reading frame (ORF), while second site deletion, either encompassing or not encompassing the pathogenic MT, mostly being flanked by short regions of DNA sequence microhomology, generates an alternative ORF with a naive stop codon, leading to a restoration of gene function.

**Figure 6 cancers-14-02504-f006:**
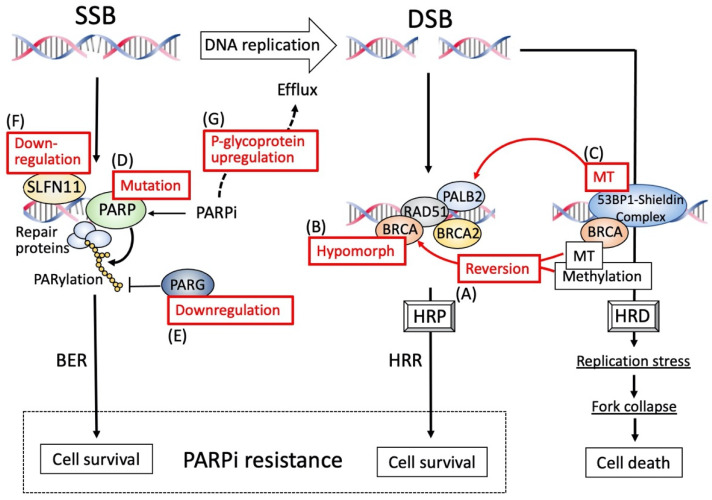
Various factors that involve PARPi resistance. PARPi resistance is conferred by proper repair of single strand breaks (SSBs) via PARP-dependent base excision repair (BER) or double strand breaks (DSBs) via homologous recombination repair (HRR), whereas DSBs cannot be repaired in the presence of the 53BP-Shieldin complex, which sequesters the PALB2-BRCA2-RAD51 complex, alternative repair molecules for HRR, conferring the HRD phenotype, leading to replication stress-induced fork collapse and cell death. Of various factors involved in PARPi resistance, seven representative causes are shown by numbers in red boxes that function to support the BER or HRR pathway. The acronyms are defined as follows: HRD, homologous recombination deficiency; HRP, homologous recombination proficiency; PALB2, partner and localizer of BRCA2; PARG, poly (ADP-ribose) glycohydrolase; RAD51, RAD51 recombinase; SLFN11, Schlafen family member 11; 53BP1, p53-binding protein 1.

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
