# Peer review of "Clinical Landscape of PARP Inhibitors in Ovarian Cancer: Molecular Mechanisms and Clues to Overcome Resistance"

_cancers, 2022, doi:10.3390/cancers14102504_

Round 1

Reviewer 1 Report

Kyo et al. nicely summarize the molecular mechanisms underlying ovarian cancer cell killing by PARPi and all known resistance mechanisms. Though this review is very thorough and focuses on ovarian cancer, I think the readers would appreciate it if a paragraph was added to the end briefly discussing PARPis in other cancers, including cancer of the breast, pancreas, etc. Discussion on other cancers need not be extensive but should include recent references. In addition, emerging therapies that synergize with PARPi should be discussed briefly as future directions. See articles below for potentially synergistic combinations: 

https://pubmed.ncbi.nlm.nih.gov/32873698/

https://pubmed.ncbi.nlm.nih.gov/33716297/

Minor: Current formatting of the manuscript merges figure legends with text. This should be fixed.

Author Response

I greatly appreciate to the reviewer for his (her) critical comments.  I basically agree with the comment that paragraphs should be added to the end briefly discussing PARPis in other cancers, including cancer of the breast, pancreas, etc.   I tried such addition, but I finally gave up because the aspects of PARPi significantly varied among cancer types, requiring considerable volume of references and paragraphs.  Since this review manuscript has already contains too many references and information at present, I feel that it is difficult to include them.  

The reviewer suggests to introduce emerging therapies that synergize with PARPi. I agree and added this information (lines 264 to 289), mainly with the concept that various approaches that target DDRs including ATM-inhibition or inhibitors can synergize with PARPi, according to the suggested manuscript that is now included as the new reference (45).

The format of figure legends are improved, being distinct from the main body of the text.

Reviewer 2 Report

The submitted review entitled "Clinical landscape of PARP inhibitors in ovarian cancer: Molecular mechanisms and clues to overcome resistance" is interesting and well written. Thanks for the privilege of reviewing your work. I enjoyed reading this manuscript and recommend it for publication.  My only criticism are that 1) there are so many abbreviations that they are hard to keep track of and somewhat distracting and 2) the figure legends are not well differentiated from the text.

Author Response

I am very sorry for so many abbreviations included in the text.  However, I think that this is a rule of the journal to show the repeated words as abbreviations.  If this is wrong, I am willing to reword them, for which  I am awaiting for the editor's opinion.  The format of figure legends are now revised, being distinct form the main body of the text.